# Dual effects of NV-CoV-2 biomimetic polymer: An antiviral regimen against COVID-19

**Ashok Chakraborty**[1]\*, **Anil Diwan**[2], **Vijetha Chiniga**[1], **Vinod Arora**[1], **Preetam Holkar**[1], **Yogesh Thakur**[1], **Jay Tatake**[1], **Randall Barton**[2], **Neelam Holkar**[1], **Rajesh Pandey**[1], **Bethany Pond**[1]

**1** AllExcel, Inc., West Haven, CT, United States of America, **2** Nanoviricides, Inc., Shelton, CT, United States of America

\* ashok.chakraborty@allexcel.com

**Data Availability Statement:** All relevant data are within the paper and its Supporting Information.

**Funding:** Funding's from Nanoviricide, Inc. However, the funders had no role in study design,

## Abstract

Remdesivir (RDV) is the only antiviral drug approved for COVID-19 therapy by the FDA. Another drug LAGEVRIO™ (molnupiravir) though has not been approved yet by FDA but has been authorized on December 23, 2021, for emergency use to treat adults with mild-to moderate COVID-19 symptoms and for whom alternative COVID-19 treatment options are not clinically appropriate. The fact is that the efficacy of RDV is, however, limited *in vivo* though it is highly promising *in vitro* against SARS-CoV-2 virus. In this paper we are focusing on the action mechanism of RDV and how it can be improved *in vivo*. The stability of RDV alone and on encapsulation with our platform technology based polymer NV-387 (NV-CoV-2), were compared in presence of plasma *in vitro* and *in vivo*. Furthermore, a non-clinical pharmacology study of NV-CoV-2 (Polymer) and NV CoV-2 (Polymer encapsulated *Remdesivir*) in both NL-63 infected and uninfected rats was done. In addition, the antiviral activity of NV-CoV-2 and NV-CoV-2-R was compared with RDV in a cell culture study. The results are (i) NV-CoV-2 polymer encapsulation protects RDV from plasma-mediated catabolism in both *in vitro* and *in vivo*, studies; (ii) Body weight measurements of the normal (uninfected) rats after administration of the test materials (NV-CoV-2 and NV-CoV-2-R) showed no toxic effects. (iii) Body weight measurements and survival rates of the NL-63 infected rats were similar to the uninfected rats after treatment with NV-CoV-2 and NV-CoV-2-R. Overall, the efficacy as an antiviral regimens were found in this order as below; NV-CoV-2-R > NV-CoV-2 > RDV. Our platform technology based NV-387-encapsulated-RDV (NV-CoV-2-R) drug has a dual effect against different variants of the coronaviruses. First, NV-CoV-2 is an antiviral regimen. Secondly, RDV is protected from plasma-mediated degradation in transit. All together, NV-CoV-2-R is the safest and efficient regimen against COVID-19.

## Introduction

So far, there are seven coronaviruses, identified that infects humans. Only 4 of the 7 belong to the beta family of coronavirus (HCoV-HKU1, SARS-CoV-2, MERS-CoV and SARS-CoV) [1]. SARS family is known to cause severe respiratory problems and disease in humans. In fact, SARS-CoV-2 infection caused a pandemic of COVID-19 that started in 2020 and caused both

data collection and analysis, decision to publish, or preparation of the manuscript.

**Competing interests:** The authors have declared that no competing interests exist.

**Abbreviations:** CoV, Coronavirus; HCoV, Human Coronavirus; FDA, Food and Drug Administration; hAPN, Human Aminopeptidase N; HAT, Human Airway Trypsin-like protease; SARS, Severe Acute Respiratory Syndrome; MERS, Middle East Respiratory Syndrome; RBD, Receptor Binding Domain; hACE2, Human Angiotensin Converting enzyme 2; RNA, Ribonucleic Acid; hDPP4, human Dipeptidyl Peptidase 4; NV-387, Nanoviricides-Polymer 387; NV-387-R, Nanoviricides-Polymer 387-Remdesivir Conjugate; RPL, Rat Plasma; SBECD, Commercial Encapsulating Agent SBECD (GILEAD); IBV, Infectious Bronchitis Virus; TMPRSSII, Transmembrane Protease; SD, Standard Deviation; SEM, Standard Error of the Mean; SBECD-R, Commercial Remdesivir Conjugated with SBECD; ADME, Absorption, Distribution, and Metabolism and Excretion.

high morbidity and mortality [2]. On 22 May 2020, there were 4,995,996 confirmed cases of SARS-CoV-2 including 327,821 deaths in 216 countries. The USA has reported the highest number of cases (1,525,186) and deaths (91,527) (mortality rate >12%) [3], and the number is increasing worldwide [2]. According to CDC reports, 94% of COVID deaths came from people with 2 to 3 comorbidities. 75% came from people with atleast 4 comorbidities [4]. *Nanoviricide* is a platform-technology-derived biomimetic polymer that can bind to virus-specific ligands [5]. The flexible polymer backbone is comprised of a polyethylene glycol (PEG) and alkyl pendants. Different virus-specific ligands can be conjugated to this PEG-based polymer backbone to produce different individual drug substances that specifically target different viral pathogens.

Viruses bind to specific cell surface ligands in order to enter the cells and infect. A *nanoviricide* is designed to act like a decoy human cell. When the virus sees the specific ligand displayed on a *nanoviricide* micelle, it is believed to bind. The flexible backbone allows the *nanoviricide* to wrap around the virus because multiple surface ligands will bind across the surface of the virus particle. As a result, this potentially leads to fusion of the *nanoviricide* with the lipid-coated virus surface through phase-inversion. It's believed that the virus is engulfed because the fatty core of the *nanoviricide* merges with the viral lipid coat and the hydrophilic shell of the *nanoviricide* becomes the exterior [6–8]. In this process, the coat proteins on the virus becomes unavailable to bind to other cells. The loss of virus particle integrity from the unavailable coat proteins will cause the virus non-infectious [9, 10].

Our objective is to develop a broad-spectrum, pan-coronavirus drug. We designed several ligands using molecular modeling of the SARS-CoV-1 S-protein, generated nanoviricide drugs from them by covalently conjugating the ligands to our platform-technology based biopolymer backbone [9–11]. The backbone of this flexible biopolymer is composed of polyethylene glycol (PEG) and some alkyl pendants. PEG will form the hydrophilic shell and render nonimmunogenicity. The alkyl chains make up the flexible core. The resulting material is different from liposomes, which are dynamic micelles. Polymeric chemical groups in our naonovircide are uniformly distributed allowing them to attach to virus-specific ligands such as peptides, antibody fragments, and various chemical moieties [12–14]. We have tested them in cell culture assays against distinctly different coronaviruses. The successful candidates were then promoted to animal studies for efficacy as well as for safety. Eventually, NV-387 is the code name for our active pharmaceutical biopolymer ingredient, and the corresponding formulated drug product is called NV-CoV-2. In this paper, both designations NV-387 and NV-CoV-2 are generally used interchangeably.

There are at least four commonly circulating coronaviruses that cause common colds, and in particular, one of these, h-CoV-NL63, binds to the same ACE2 receptor as SARS-CoV-1 and SARS-CoV-2, but cause less severe lung pathology and morbidity with significantly lower fatality rates [1, 15–18]. We developed an animal model based on hCoV-NL63 lethal lung infection that mimicked the SARS-CoV-2 lung pathology in humans to test effectiveness of our drug candidates during the screening process.

We will be discussing three important aspects of the drug. First, the drugs antiviral activity *in vitro* cell culture model and *in vivo* rat animal model. Secondly, its capability for increasing the half-life of RDV in presence of plasma in both *in vitro* and *in vivo* studies will also be reported. Finally, we will discuss some safety aspects of this drug.

## Methods

### Materials

Nanoviricide® is a platform technology-based biopolymer called NV-CoV-2, that is used as a broadspectrum antiviral compound. The theoretical molecular formula of NV-CoV-2 is

$C_{104}H_{188}N_2O_{44}S_4$. The IUPAC name of the polymer NV-CoV-2 is derived using MarvinSketch 6.2.1. Its based on 22 repeated units (m) of polyethylene glycol substituted by 2 Hexadecylamine ($R_1$), the IUPAC name of NV-CoV-2 repeat unit is 2-({1-carboxy-3-[(1-{[2-carboxy-1-(hexadecylcarbamoyl) ethyl]sulfanyl}-3-({3-carboxy-3-[(1,2-dicarboxyethyl)sulfanyl]propanoyl} oxy)-4-{[1-(hexadecylcarba-moyl)-3-[(65-hydroxy-3,6,9,12,15,18,21,24,27,30,33,36,39,42,45,48,51,54,57,60,63-henicosaoxapenta-hexacontan-1-yl)oxy]-3-oxopropyl]sulfanyl}butan-2-yl)oxy]-3-oxopropyl}sulfanyl)butanedioic acid. Based on polyethylene glycol of 22 repeat units (m) substituted by two hexadecylamine (R1), the theo-retical molecular formula for the repeat unit of the polymeric drug substance (NV-CoV-2, 6) is C104H188N2O44S4. Nominal calculated formula weight of the polymer repeat unit (RU) is 2298.85 g/mol when the hexadecylamine substitution level (x) per repeat unit is 2, the MMSA substitution level (y) is also 2, and the starting polyethylene glycol (P10, 1) consists of 22 repeat units (m). The actual substitution levels are less than these theoretical limits and are described in the drug substance specifications. The repeat unit (RU) formula weight at hexadecylamine (HDA) substitution level x between 0 and 2 is described by the formula: MWRU = 1853 + 223x. The degree of polymerization, n, in P10M2DT (HDA)x (MMSA)y polymer is 8 ± 2 [12, 14].

NV-CoV-2 appears to be an off-white, waxy, non-crystalline semi-solid at room tempera-ture. The drug substance has a slight, sour taste but there is no noticeable smell [12]. Pharma-ceutical properties, formulations for injection, physical properties, and chemical properties are all available elsewhere [12–14].

**Other reagents.** RDV, GS-441524 and their internal standard ($^{13}C_6$- and $^{13}C_5$-isotopes) were procured from Medko Biosciences, Inc. (USA) and AlsaCHIM (USA), respectively.

**Cells and viruses.** Rhesus Monkey Kidney Epithelial (LLC-MK2) cells and Human Fetal Lung Fibroblast (MRC-5) cells were obtained from ATCC and grown in Dulbecco's Modified Eagle Medium (DMEM), supplemented with 10% fetal bovine serum (FBS), and 1% Penicillin-Streptomycin-Amphotericin B according to provider's instructions. Human Coronavirus 229E (HCoV-229E) and NL63 (HCoV-NL63) were purchased from ATCC and propagated on MRC-5 and LLC-MK2 cells, respectively at 34˚C for 5–7 days according to manufacturer's instructions to produce virus stocks. These virus stocks were aliquoted and stored at -80˚C until needed.

**Virus titer.** Coronaviruses titers were determined using virus plaque assay [19–22]. Briefly, MRC5 cells (for HCoV-229E) and LLC-MK2 (for CoV-NL63) were grown in 6 well plates in duplicate at density of 200,000 cells per ml. Two hundred μl of each diluted virus was incubated with the cells for an hour at 34C before adding 2X DMEM medium mixed with 4% low melting agarose to each well. Plates left for 1 hour to allow the agarose plug solidification before incubation at 34C for 5-7days (for CoV-229E) 6–9 days (for CoV-NL63). Cells were fixed with formalin and stained with 2% crystal violet.

**Compound and virus dilutions.** LLC-MK2 cells (for CoV-NL63 virus) and MRC-5 cells (for CoV-229E virus) were plated in 96-well black, clear bottom plates at a density of 30,000 cells per well in 100 μl media for 24hr prior to infection. Stock compounds were made at a con-centration of 20 mg/ml in various vehicles. On the day of infection, compounds were serially diluted in DMEM without serum to 4x the required dose. In a separate clear 96-well plate 60 μl of compound were added to 60 μl of virus and incubated for 1 hour at 34˚C. After incubation, 100 μl/well of compound and virus mixture was added to each well of the cell plates, for a total of 200 μl and 1x dilution of compound. Plates were incubated at 34˚C for appropriate amount of time for each virus (CoV-229E or CoV-NL63, for 5 days or 7 days, respectively, for CPE).

**Cytotoxicity/cell-viability assay.** We used the Promega CellTiter-Glo luminescent cell viability assay to test for compounds cytotoxicity. This assay allows the differentiation between death caused by compound toxicity and actual virus suppression/cytopathic effect. The assay was performed according to manufacturer's instructions [23].

**Virus Cytopathic Effect (CPE) assay.** A cell-based viral CPE assay in 96-well black, clear-bottom plates was developed which allows a higher-throughput screening of compounds. Virus (CoV-229E and CoV-NL63) were pre-incubated with compounds and then added to MRC-5 cells or LLC-MK2 cells, respectively, and incubated for 5–7 days. This gives the virus time to replicate and cause cell death (cytopathic effect). The Promega CellTiter-Glo Luminescent Cell Viability Assay was used to measure cell viability and determine the amount of cell death (CPE) caused by the virus [24]. Relative CPE levels can be compared between all groups to identify any reduction in viral replication, as a reduction in CPE compared to untreated infected controls suggests a decrease in virus production/growth/spread.

*Quantitative composition of drug product NV-CoV-2 solution for injection.* The drug product NV-CoV-2 (50 mg/ml) solution for injection will be supplied in a single vial as 10 ml. D-mannitol (40 mg/ml) was added to maintain a desired osmolality. This solution was prepared to deliver 450 mg of NV-CoV-2 drug substance in a 9 ml injection volume [25, 26].

*Pharmacokinetics.* Conventional methodologies that measure the pharmacokinetic properties of small molecules are not suitable for NV-CoV-2 because it is composed of a flexible PEG- based polymer backbone. Instead, direct ELISA was developed to quantitatively detect NV-CoV-2 in biological matrices in order to measure its pharmacokinetic properties [26].

We conducted a non-GLP study to determine the levels of systemic exposure of NV-CoV-2 when NV-CoV-2 drug is administered to rats once a day for 5 days (0,1,3,5 and 7) over an 8-day period. Direct ELISA was used for detection and quantification of NV-CoV-2 in all the plasma samples. This assay was a colorimetric ELISA, which measured the levels of NV-CoV-2 in the processed plasma samples [27]. Thirty-six animals (three/sex/group) were administered NV-CoV-2, at three different doses (Table 1).

The vehicle and NV-CoV-2 drug was administered once a day, over an eight-day period on days 0, 1, 3, 5 and 7. Day 0 represented the 1st injection and Day 7 was the 5th injection. Each compound was delivered through a slow-push IV injection. Blood samples for the systemic exposure were collected at 0, 0.08, 0.5, 1, 2, 4, 8 and 24 hours after the 1st and 5th injection of drug administration. The blood samples were taken from one male and one female rat in each test article treated group at all time points. The same animal was used for the blood samples throughout all the time points of the study from Day 0 and Day 7 injections. A direct ELISA was done to detect and quantify NV-CoV-2 in all the plasma samples. 5G8-Biotin, also known as biotinylated detection antibody, was used because it is specific to NV-CoV-2. The direct ELISA for NV-CoV-2 was previously characterized for linearity and detection limit, where the limit of detection (LOD) is 0.25 μg/ml.

*In vitro antiviral activity of NV-CoV-2.* The antiviral efficacy and potency, and the cytotoxicity of the antiviral compound NV-CoV-2 in cell cultures were studied. Since HCoV-NL63 virus, like SARS-CoV-2, binds to angiotensin-converting enzyme 2 (ACE2) for entry into cells we have used CoV-NL63 as a better surrogate virus for studying SARS-CoV-2 *in vitro* and *in vivo* [17, 28]. In brief, LLC-MK2 cells (for CoV-NL63 virus) and MRC-5 cells (for CoV-229E

**Table 1. Treatment groups and regimen.**

| Treatment Group and Regimen | Concentration (mg/mL) | Dosage (mg/kg) | Total Dosage (mg/kg) | # of Rats | |
|---|---|---|---|---|---|
| | | | | M | F |
| Untreated | 0 | 0 | 0 | 5 | 4 |
| Vehicle *i.v.* (Days 0,1,3,5 and 7) | 0 | 0 | 0 | 8 | 8 |
| Remdesivir *i.v.* (Days 0,1,2,3,4,5,6 and 7) | 1 | 10 | 90 | 8 | 8 |
| NV-CoV-2 Med *i.v.* (Days 0,1,3,5, and 7) | 16 | 160 | 800 | 8 | 8 |
| NV-CoV-2 High *i.v.* (Days 0,1,3,5 and 7) | 32 | 320 | 1600 | 8 | 8 |

virus) were plated in 96-well black, clear bottom plates at a density of 30,000 cells per well in 100 μl media for 24hr prior to infection.

Stock compounds were made at a concentration of 20 mg/ml in various vehicles. On the day of infection, compounds were serially diluted in DMEM medium 4x the required dose (i.e., a final concentration of 320 μg/ml was initially made at 1280 μg/ml). Virus was then diluted to the appropriate infection PFU/well dose: CoV-NL63 and CoV-229E MOI = 0.01. In a separate 96-well plate 60 μl of compound were added to 60 μl of virus and incubated for 1 hour at 34˚C. After incubation, 100 μl/well of compound: virus mixture was added to each well of the cell plates, for a total of 200 μl and 1x dilution of compound. Plates were incubated at 37˚C for appropriate amount of time for each virus and subjected to CellTiter-Glo assay to measure the intracellular level of ATP. As a control, PBS was used instead of test compounds. Cells were incubated for 1 hour at 37˚C $CO_2$ incubator. After that cells were washed with PBS, and fresh DMEM medium supplemented with 2% FBS was added to the cells. Cells were incubated 24 hours at 37˚C.

We used the Promega CellTiter-Glo luminescent cell viability assay to test for compounds cytotoxicity [29]. This assay allows the differentiation between death caused by compound toxicity and actual virus suppression/cytopathic effect (CPE). Media were removed, and cells were washed with PBS before adding 100 μl of serum-free media to each well. The assay was performed according to manufacturer's instructions. The concentration of compound that causes reduction of cell viability by 50% is termed Effective dose or concentration 50 (EC50).

### *In vivo* antiviral activity of NV-CoV-2

*Dose-response of NV-CoV-2 in rats infected with CoV-NL63. Animal protocol.* The entire animal handling experiments were done by Dr. Krishna Menon from AR Biosystems, (17633 Gunn Highway, Odessa, FL 33556), based on the protocol #IACUC No. 14/17ARB. This study followed strict guidelines and recommendations from the "Guide for the Care and Use of Laboratory Animals of the National Institutes of Health." All surgeries were performed under sodium pentobarbital anesthesia in order to minimize suffering. There aren't any known acceptable alternatives to the use of live animals to accomplish the purpose of this study. The large historical database concludes that animal study is a standard model for non-clinical studies. In fact, previous investigations with the same disease models found this to be the most suitable model. The rats were observed daily for body weight, clinical and behavior changes. At the end of the experiment, the rats were sacrificed by $CO_2$ asphyxiation. The primary end points were survival and loss of body weight. This study was designed to use the fewest amount of animals possible. The amount of animals used was consistent with the scientific objectives of the study, the contemporary scientific standards and for applicable regulatory requirements. The number of animals used is considered the minimum necessary for meaningful biological comparisons and statistical calculations.

Both male and female Sprague Dawley rats, all 8 to 9 weeks old, were infected with $10^4$ CoV-NL63 viral particles directly into the lungs. The untreated rats were infected in the same manner actually succumb to the disease in 5 to 6 days. This group of rats was used as a model for evaluating the efficacy of NV-CoV-2. The drug, NV-CoV-2 and vehicle was injected daily through a tail vein IV (10 ml/kg) once a day on days 0, 1, 3, 5 and 7 over a 7-day period. RDV was also injected through a tail vein IV twice on the first day followed by once a day for the remaining 7 days. A group of infected, untreated rats was included as an additional control. The groups of animals and the treatment protocols are shown in Table 1.

*Effect of number of days of NV-CoV-2 treatment in rats infected with CoV-NL63.* 8 to 9 weeks old male and female Sprague-Dawley rats were infected with 2 x $10^4$ CoV-NL63 viral

**Table 2. The groups of animals and the treatment protocol.**

| Group | NV-CoV-2 (mg/ml) | Day of Injection | No of Injections | Injection Volume (mL/kg) | NV-CoV-2 Dose/ Injection (mg/kg) | Total NV-CoV-2 Injected (mg/kg) | No of Animals | |
|---|---|---|---|---|---|---|---|---|
| | | | | | | | M | F |
| NV-CoV-2 1 Day | 50 | 0 | 1 | 12 | 600 | 600 | 7 | 7 |
| NV-CoV-2 3 Days | 50 | 0, 1 and 3 | 3 | 12 | 600 | 1,800 | 7 | 7 |
| NV-CoV-2 5 Days | 50 | 0, 1, 3, 5 and 7 | 5 | 12 | 600 | 3,000 | 7 | 7 |
| Vehicle Control | 0 | 0, 1, 3, 5 and 7 | 5 | 12 | 0 | 0 | 4 | 4 |
| Infected Untreated | | | 0 | 0 | 0 | 0 | 2 | 2 |

particles directly into the lungs. All untreated rats infected in the same manner succumb to the disease within 5 to 6 days. Therefore, this group of rats was used as a model for evaluating the efficacy of NV-CoV-2. Depending on the type of treatment the groups of rats was receiving determined the different number of days for that particular treatment. NV-CoV-2 was administered through a tail IV injection, once a day on days 1, 3 and/or 5. The vehicle control treated rats were treated for 5 days on Days 0, 1, 3, 5 and 7. A group of infected, untreated rats was included as an additional control. The groups of animals and the different treatments are shown in Table 2 below.

The infected rats were treated with NV-CoV-2 at the same dose per injection of 600 mg/kg. The only variable that changed in this study was the number of days the treatment occurred. The rats were observed daily for changes in their body weight and clinical and behavior changes. The moribund rats were sacrificed. In the end, the primary end points were survival and body weight loss.

*Protection efficiency of NV-CoV-2 encapsulation of RDV (NV-CoV-2-R) from plasma-mediated degradation in vitro.* A standard curve of RDV was determined using LC-MS at different concentrations ranging from 0–5 ng/μL in DMSO:MeOH (1:9). As an internal standard (ISTD), $^{13}C_6$-ISTD (2.5 ng/μL) was used as well. The extraction of RDV was completed in the LC-MS assay by using acetonitrile. The concentrations of RDV were compared to their ratio with the ISTDs ($^{13}C_6$-ISTD) in order to generate the standard curve for RDV.

Secondly, 10 μL of the test materials (20 ng/μL) was incubated with 30 μL of rat plasma at different time points (as indicated in the Result section) and extracted with acetonitrile (100 μL). The mixture was centrifuged for 10 minutes at 10,000 x g to separate the supernatants from the precipitates. The supernatants contained RDV, which was determined by LC-MS chromatography [12].

The ratio of RDV and its isotope, $^{13}C_6$-ISTD (as an internal standard), was calculated. The amount of RDV was determined using the linear equation derived from their respective standard curve. The values were then normalized using the dilution factor for the original plasma samples.

*Survival rate of NL-63 infected male rats after treatment with NV-CoV-2 encapsulated RDV (NV-CoV-2-R).* There were 36 male Sprague Dawley rats (Taconic Biosciences, USA) (three/ in control and in treatment groups) administered with NV-CoV-2 or NV-CoV-2-R. Treatment was once a day for 5 days (0, 1, 3, 5, and 7) over a 7-day time period. NV-376 (RDV-in-SBECD; Gilead) was given as two doses on day 1 followed by a daily dose until day 7. DMSO was used as a vehicle for the control group. Injection on day 0 is considered as a 1st injection and on day 7 is the 5th injection. Each compound was delivered via slow-push IV injection.

Blood samples for systemic exposure assay were collected at 0, 0.08, 0.5, 1, 2, 4, 8 and 24 hours after 1st and 5th injection of the drugs.

Blood samples were taken from one animal in each group from all the time points. The same animal was used to collect blood samples throughout the entire study at all the time points after "day 0" and "day 7" injections. All the detailed procedures and the study design are explained in more depth previously [30]. RDV values obtained in the rat plasma (mg/mL) after the 1st. and 5th injection of the drugs was normalized by dividing with the amount of RDV administered (mg/kg of rat body weight).

**Safety pharmacology studies.**   We have conducted safety pharmacology studies. The core battery tests were defined in the ICH S7A guidelines on the respiratory and central nervous systems in the rat and the cardiovascular system in the monkey after IV administration of NV-COV-2.

**Respiratory system and central nervous system.**   There was an evaluation of pulmonary and neurobehavioral functions following intravenous administration of NV-CoV-2 in 24 conscious male rats (Calvert Laboratories, Inc., Scott Township, PA) (Strain/Substrain:Crl:CD®(SD)]. The rats were up to 6 weeks old and ranged between162-199 in their body weight. Each group contains 6 animals (4 groups total; Vehicle, NV-CoV-2 Low (25 mg/kg); NV-CoV-2 Medium (50 mg/kg); NV-CoV-2 high (100 mg/kg). The drug (5ml/kg) was injected through IV.

**Pulmonary evaluation.**   Prior to the experiment, the rats were trained for two days in the head-out plethysmograph chamber. Each animal of each test group was weighed and placed in the plethysmograph chamber on the day of treatment (SOP PHA-EQP-96). The baseline respiratory parameters were obtained for 5 minutes, followed by a stabilization period. Each rat was removed from the chamber and dosed based on their assigned test groups. After the treatment dose was administered, the rat was returned to its designated plethysmograph chamber and allowed to stabilize. The respiratory parameters were recorded again but measured for 5-minute intervals at 15 min, 1 hour, 2 hours and 4 hours time points. Respiratory parameters were measured at ±5 minutes for the 1 hour evaluation and ±15 minutes for the 1, 2 and 4 hours post-dose evaluations. After each reading, the animals were removed from the plethysmograph chamber and returned to the home cage. The following parameters were recorded (SOP PHA-SOF-90) using the Ponemah Physiology Platform (Ponemah v.5.20 Pulmonary): Respiratory rate; Tidal Volume, and Minute Volume [31].

**Neurobehavioral evaluation.**   An modified Irwin and functional observations battery tests were used for the neurobehavioral evaluation. At approximately 15 minutes, 1 hour and 4 hours after treatment dosing and the completion of the pulmonary evaluation was finished, the animals were placed in a fixed environment. The rats were separated into groups and placed in a Plexiglas® enclosure with a fitted lid. The enclosure was placed on absorbent paper that detects excretions. In this environment, the rats were free to move about and interact with the other rats.

The observations were recorded for the presence of the following symptoms of the group in the Plexiglas® enclosure. The animal number was documented when any of the following symptoms were observed. These symptoms included seizures/convulsions (awareness reaction), body tremors (motor activity), ataxia (piloerection) and abnormal posture (stereotypy). Excretion (decreased respiration) was documented separately as a group observation [32, 33].

The different tests done on all the animals were as followed. Clapping the hands immediately above the open square and watching the response and reaction tested for the startle response. Each animal was individually removed from the square and examined for vocalization, irritability and increased secretion of saliva or tears. Changes in abdominal tone were also recorded by feeling the rat's abdominal muscles. The loss of the righting reflex was examined by placing each animal on its back and monitored how they behaved. Pupil size was recorded for mydriasis or miosis under a magnifying glass. An increase or decrease in their

nociceptive response was tested by a tail pinch. A cotton ball and blunt probe checked the corneal reflex and pinnal reflex. Each animal was tested for immobility and grip strength on an inclined wire screen.

**Body temperatures.** In addition, rectal body temperatures were taken from all the animals approximately 1 and 4 hours after dosing using a YSI Precision Digital Thermometer (Model 4600).

**Cardiovascular system.** There was an evaluation of the cardiovascular function after administration of NV-CoV-2 in conscious telemetered cynomolgus monkeys (Calvert Laboratories, Inc., Scott Township, PA). There were 4 male monkeys, between the ages of 2 to 5 years, which weighed approximately 5.1 to 6.0 kg prior to treatment (Species: Monkey; Breed: Cynomolgus (*Macaca fascicularis*); Total Number /Sex): 4 Male; Age Range: 2–5 years; Body Weight Range: 5.1 to 6.0 kg prior to treatment). The test article administration and group assignments are shown below in Table 3.

NV-CoV-2 formulations were infused intravenously once using a calibrated infusion pump into the saphenous vein over a period of 30 (± 2) minutes by a percutaneous IV catheter. Animals were restrained during dosing to prevent any movement. Prior to dosing, all the animals were trained on the restrainers.

**Cardiovascular evaluation.** Data was collected for 24 hours before the first dosing day in order to establish acceptable baseline parameters for each animal. The baseline data was retained in the study file, but not in the final report. Pre-dose values were determined from data that was collected at least 24 hours prior to each subsequent dose given to the animals. Following the dose administration, parameters were recorded continuously for 24 hours. A washout period of approximately 6 days was allowed between test article doses [34].

The following parameters were analyzed using the Ponemah 5.20 SP9 and ECG Analysis Module v. 5.30 software (SOP PHA SOF-106): Arterial Blood Pressure, Heart Rate (HR), ECG, and Body Temperature [35]. One-minute tracings of the ECGs occurred 15 minutes prior to dosing. After the dosing occurred, one-minute tracing of the ECGs happened at 5 minutes, 30 minutes, 1 hour, 2 hours, 4 hours, 8 hours, 12 hours and 24 hours. Any morphological changes that occurred in the ECG printouts were evaluated by a board-certified veterinary cardiologist.

**Immunogenicity.** Since NV-CoV-2 was administered through an IV injection, it is important to determine if there is any evidence of immunogenicity. A study was conducted for detecting any anti-NV-CoV-2 antibodies in the rat serum samples. These serum samples were collected on day 14 and 28. NV-CoV-2 was given to the animals in 6 sequential IV injections over a 10-day period. In order to quantify and detect any anti-NV-CoV-2 antibodies in the serum samples, an Indirect ELISA assay was preformed [36, 37]. This assay is a colorimetric method designed to measure the levels of anti-NV-CoV-2 in liquid samples. The antigen, NV-CoV-2, is first immobilized in each well of a 96-well assay plate. Secondly, both diluted serum samples or positive control and normal serum spiked with rat anti-NV-CoV-2 antibodies are incubated. The bound anti-NV-CoV-2 antibody is then detected by incubating with 1:1

**Table 3. The test article administration schedule (Latin square design).**

| Treatment | Dose Level (mg/kg) | Dose Conc. (mg/mL) | Day of Dosing[a] | | | |
|---|---|---|---|---|---|---|
| | | | Animal #1 | Animal #2 | Animal #3 | Animal #4 |
| Vehicle | 0 | 5 | Day 1 | Day 8 | Day 15 | Day 22 |
| NV-CoV-2 | 25 | 5 | Day 22 | Day 1 | Day 8 | Day 15 |
| NV-CoV-2 | 37.5 | 5 | Day 15 | Day 22 | Day 1 | Day 8 |
| NV-CoV-2 | 50 | 5 | Day 8 | Day 15 | Day 22 | Day 1 |

[a] A washout period of at least 3 days was allowed between each dose administration.

mixture of HRP-conjugated Goat Anti-Rat IgG and IgM antibodies. The assay was quantitated with the chromogenic HRP substrate (TMB). A purified preparation of monoclonal antibodies (IgG and IgM) that are specific to NV-CoV-2 was used as the positive control in the naive subject matrix, which determined the sensitivity. This Indirect ELISA method is specific and selective to NV-CoV-2 detection and quantification.

## Ethical statement

The entire animal handling experiments were done by Dr. Krishna Menon from AR Biosystems, (17633 Gunn Highway, Odessa, FL 33556), based on the protocol #IACUC No. 14/17ARB. This study followed strict guidelines and recommendations from the "Guide for the Care and Use of Laboratory Animals of the National Institutes of Health." All surgeries were performed under sodium pentobarbital anesthesia in order to minimize suffering. There aren't any known acceptable alternatives to the use of live animals to accomplish the purpose of this study. The large historical database concludes that animal study is a standard model for non-clinical studies. In fact, previous investigations with the same disease models found this to be the most suitable model. The rats were observed daily for body weight, clinical and behavior changes. At the end of the experiment, the rats were sacrificed by $CO_2$ asphyxiation. The primary end points were survival and loss of body weight.

This study was designed to use the fewest amount of animals possible. The amount of animals used was consistent with the scientific objectives of the study, the contemporary scientific standards and for applicable regulatory requirements. The number of animals used is considered the minimum necessary for meaningful biological comparisons and statistical calculations.

**Statistical analysis.** The t-test and one-way analysis of variance were used for statistically analyzing the data. Percentages of variation for all values are below 10%.

## Results

### Pharmacokinetics study results

The analytical method used for detection and quantification of NV-CoV-2 in plasma samples was direct ELISA. The result shows that for all doses, NV-CoV-2 was detected in rat plasma producing an initial increase that peaked between 4–8 hours and decrease below the detection level between 24 and 48 hours (Table 4).

**Table 4. Results of NV-CoV-2 pharmacokinetics analyses in groups treated with NV- CoV-2.**

| | Dose of NV-CoV-2 (mg/kg) | Dose Volume (ml/kg) | NV-CoV-2 Concentration (mg/ml) | $t_{max}$ (hrs.) | $C_{max}$ (mg/ml) | +/- SD (mg/ml) |
|---|---|---|---|---|---|---|
| NV-CoV-2-High Male-Day1 | 320 | 10 | 32 | 8 | 0.64 | 0.06 |
| NV-CoV-2-High Male-Day 7 | 320 | 10 | 32 | 0.08 | 2.60 | 0.29 |
| NV-Cov-2-Medium Male-Day1 | 160 | 10 | 16 | 24 | 1.39 | 0.13 |
| NV-CoV-2-Medium Male-Day 7 | 160 | 10 | 16 | 4 | 1.55 | 0.07 |
| NV-CoV-2-High-Female Day1 | 320 | 10 | 32 | 4 | 2.48 | 0.10 |
| NV-CoV-2-High-Female Day 7 | 320 | 10 | 32 | 4 | 1.02 | 0.03 |
| NV-CoV-2-Medium-Female Day1 | 160 | 10 | 16 | 8 | 0.50 | 0.06 |
| NV-CoV-2-Medium-Female Day 7 | 160 | 10 | 16 | 4 | 0.08 | 0.05 |

Results of NV-CoV-2 pharmacokinetics analyses in groups treated with the drug. The observed maximum plasma concentration ($C_{max}$) and the time it takes to reach $C_{max}$, ($t_{max}$) are shown above. $C_{max}$ is expressed in mg/ml and $t_{max}$ is in hours.

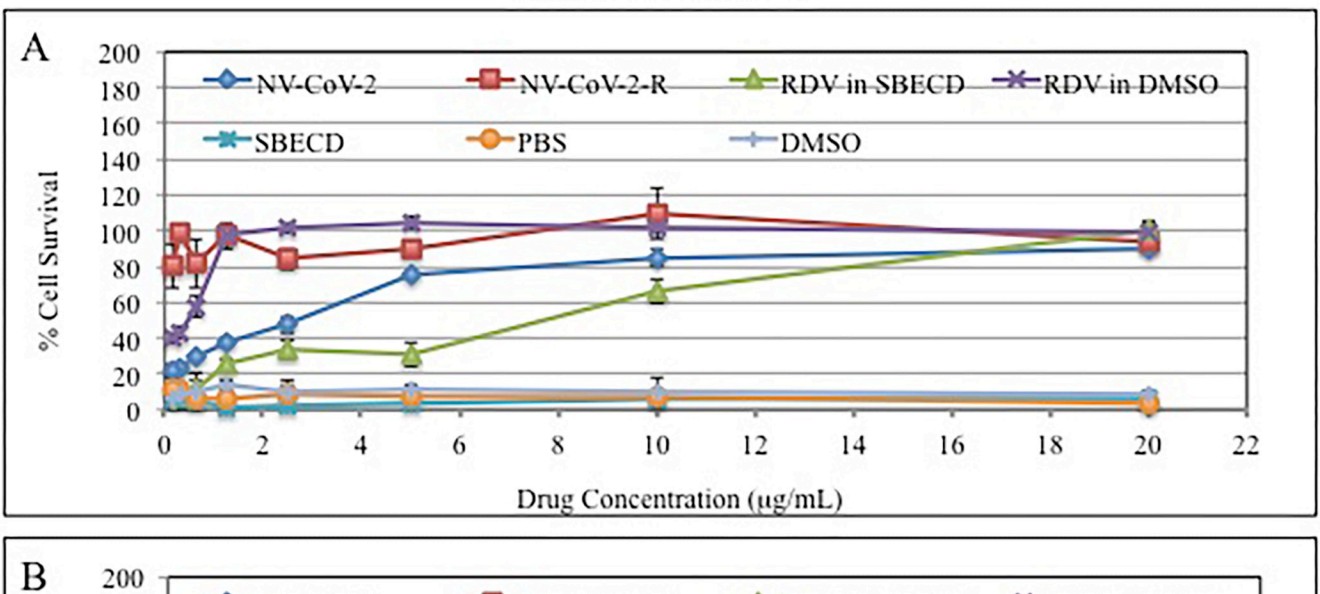

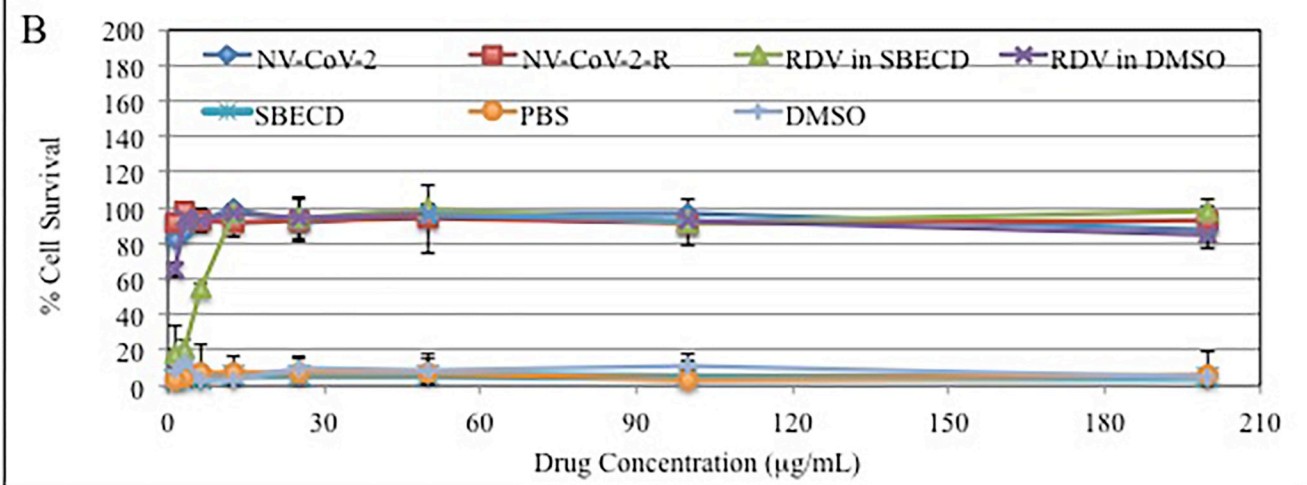

**Fig 1. Antiviral effect of NV-CoV-2, NV-CoV-2-R and RDV *in vitro*. A:** MRC-5 cells infected with CoV-229E virus; **B:** LLC-MK2 cells infected with NL63 virus.

### *In vitro* antiviral activity of NV-CoV-2 (Fig 1)

Dose-responsive antiviral effects of NV-CoV-2, NV-CoV-2-R and RDV were observed as a protection of MRC-5 cells or LLC-MK2 cells infected with CoV-229E virus (A) or CoV-NL63 (B) virus, respectively. Antiviral efficacy is in the following order with MRC-5 cells: RDV $\leq$ NV-CoV-2-R < NV-CoV-2. For the cells, LLC-MK2, the order of antiviral capacity are: RDV = NV-CoV-2-R = NV-CoV-2. PBS and DMSO, as a negative control do not show any antiviral capacity in either case.

### *In vivo* antiviral activity of NV-CoV-2

**Dose-response of NV-CoV-2 in rats infected with CoV-NL63 (Fig 2A).** The untreated and vehicle-treated rats infected with the CoV-NL63 virus survived up to day 5. Rats treated daily with RDV (10 mg/kg) survived 7.5 days. The rats treated with NV-CoV-2 (160 and 320 mg/kg) survived until 13.5 and 14 days. Therefore, treatment with NV-CoV-2 extended the

## A: Dose-Response

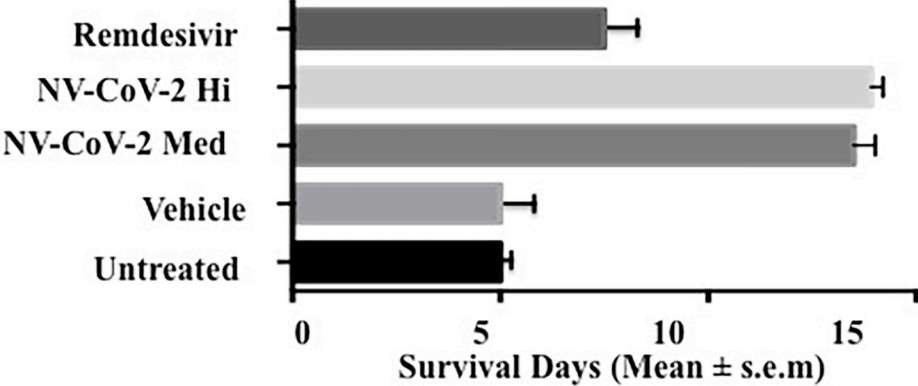

## B: Effect of Number of Days of Treatment

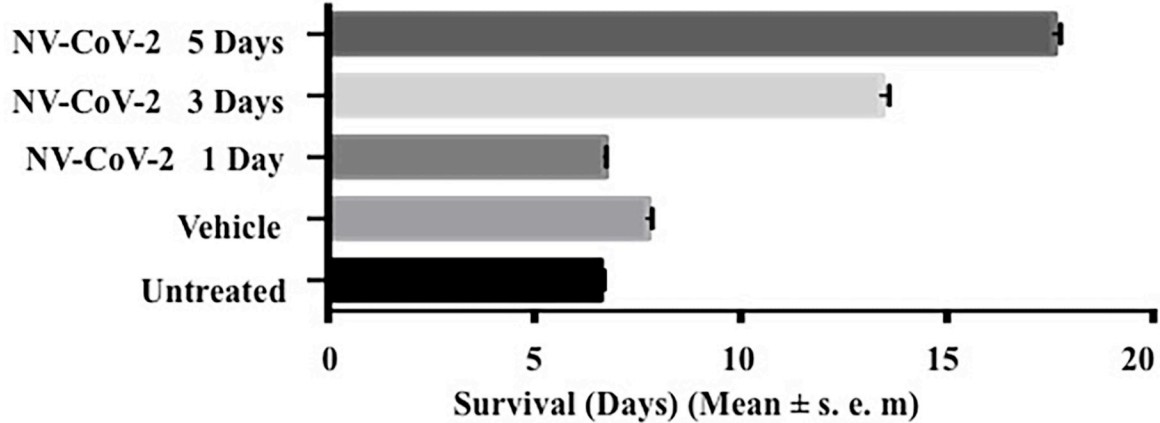

**Fig 2. Effect of NV-CoV-2 in rats infected with CoV-NL63, *in vivo*.**

survival of rats infected intra-tracheally from the lethal dose of the human CoV-NL63 virus. More importantly, NV-CoV-2 treatment was better than RDV treatment alone.

**Effect of number of days dosing of NV-CoV-2 in rats infected with CoV-NL63 (Fig 2B).** The survival of rats with the lethal dose of CoV-NL63 was dependent on the number of days of the drug used for the treatment. An increase in the number of days of treatment correlated with an increase in the survival days. The survival of the rats administered with NV-CoV-2 for 1 day was similar to the untreated rats and vehicle-treated rats of 6–7 days. When the treatment injection was for 3 to 5 days, the survival increased to 13.4 to 17.6 days.

**Protection efficiency of NV-CoV-2 encapsulation of RDV (NV-CoV-2-R) from plasma mediated degradation *in vitro* and *in vivo*.** NV-CoV-2 is capable of encapsulating other drugs [9, 10]. This ability was exploited by creating another drug candidate, NV-CoV-2-R, which encapsulates inside NV-CoV-2 the only approved antiviral drug against SARS-CoV-2, RDV [25, 26]. *In vitro*, NV-CoV-2 polymer encapsulation protects RDV from plasma-mediated catabolism (Fig 3). *In vivo* study with male rats animal model, the results were shown in Fig 4. After the 1st injection of NV-CoV-2-R-Med, the accumulation of RDV is greater which is more evident after 5th injections. No such differences were found in the accumulation of

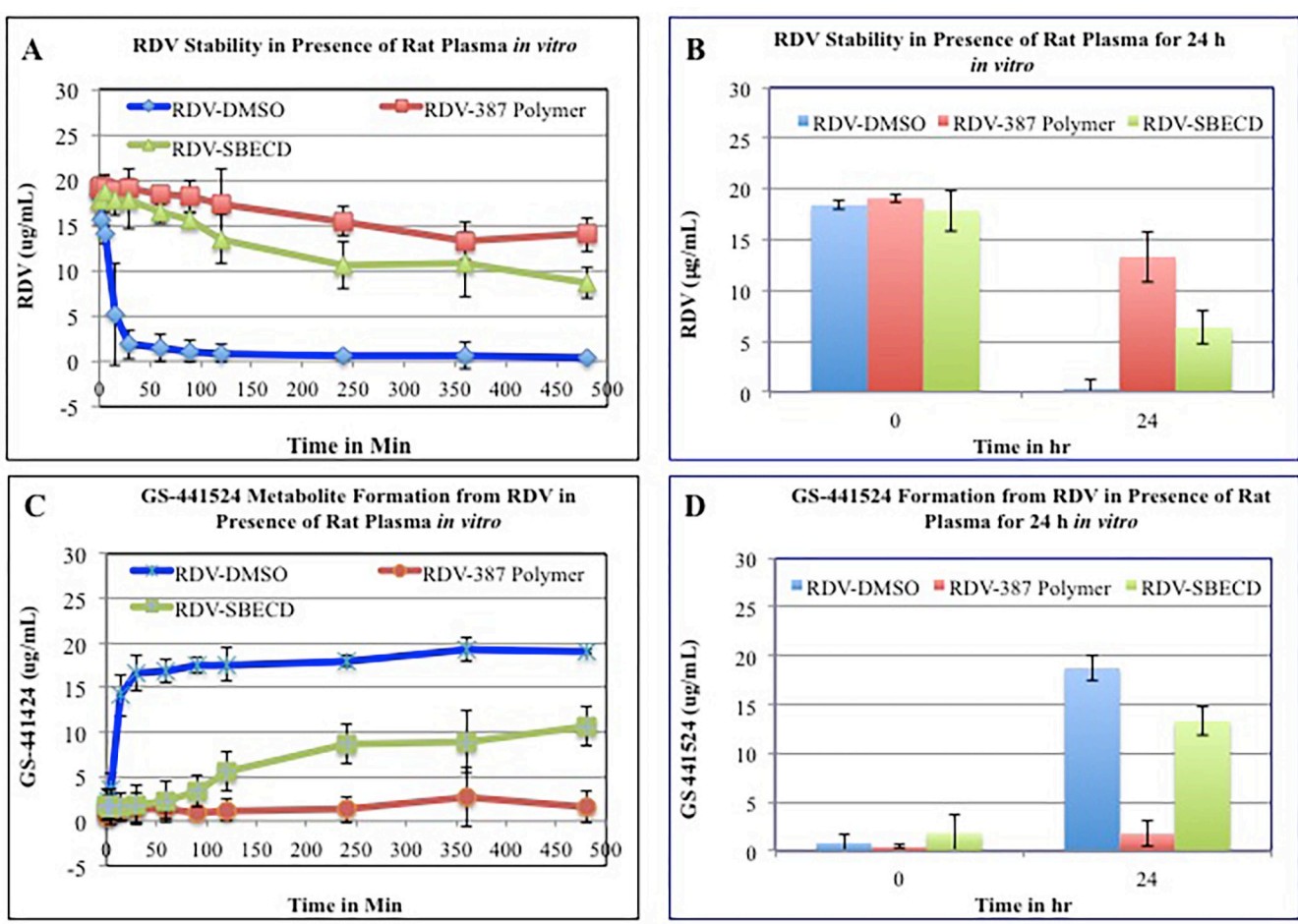

**Fig 3. Protection efficiency of NV-CoV-2 encapsulation of RDV (NV-CoV-2-R) from plasma mediated degradation *in vitro*.**

RDV from 376-R-SBECD (Gilead) injections, both 1st and 5th. In brief, NV-CoV-2 polymer can encapsulate *Remdesivir* efficiently.

**Survival rate of NL-63 infected male rats after treatment with NV-CoV-2 encapsulated RDV (NV-CoV-2-R).** The group of rats treated with NV-CoV-2-R had a slight increase in survival after 1 day of treatment of 8.8 days. The vehicle-treated rats only survived 7.7 days. When NV-CoV-2-R injection was for 3 to 5 days, the survival increased to 13.9 and 18 days (Fig 5). Similarly to NV-CoV-2 treatment, the survival increased after 3 to 5 days of NV-CoV-2-R treatment. It was reflected in the body weight changes over time after infection.

## Safety pharmacology studies

**Body weight measurement.** The efficacy of NV-CoV-2 on the survival was also reflected in the body weight changes. Loss of body weight increased significantly as the animals became moribund. By day 5 after infection, the infected rats, untreated and vehicle-treated rats loss 30.8 and 26 g in body weight. The loss in body weight in all other groups ranged between 3.0 and 4.5 g. On Day 7, RDV-treated rats lost 30.1g in body weight, whereas the loss in body weight of both NV-CoV-2- and NV-CoV-2-R-treated rats was not significantly different from day 5. By day 13, both medium (160 mg/kg/ injection) and high dosed (320 mg/kg/injection) NV-CoV-2 treated rats significantly lost 25.7 and 12.6 g in body weight.

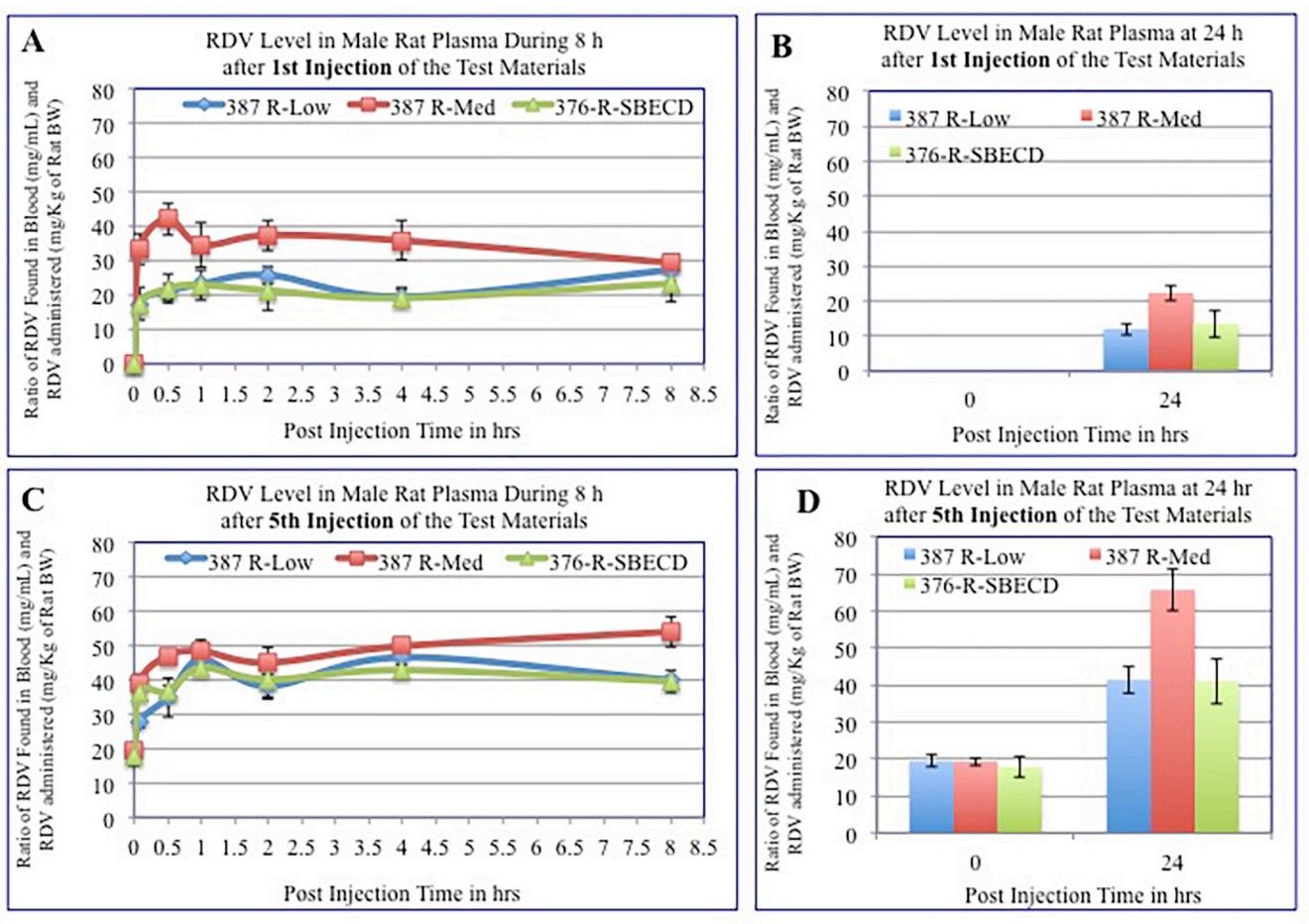

**Fig 4. Protection efficiency of NV-CoV-2 encapsulation of RDV (NV-CoV-2-R) from plasma mediated degradation *in vivo*.**

As a result, the treatment with NV-CoV-2 for both 160 and 320 mg/kg/injection (Medium and High doses, respectively) on days 0, 1, 3, 5 and 7 had significant effect on the survival in the rats infected with the human coronavirus CoV-NL63. Therefore, NV-CoV-2 treatment against the human coronavirus infection is clearly the superior treatment than RDV.

**Respiratory parameter results.** Effects of the vehicle and NV-CoV-2 (25, 50 and 100 mg/kg) on pulmonary parameters are shown in the Table 5 below.

The intravenous administration of NV-CoV-2 at 25, 50 or 100 mg/kg did not induce any significant effect on respiratory rate, tidal volume or minute volume when compared to the vehicle control group.

**Central nervous system parameter results.** As shown in the Table 6 below, no apparent neurobehavioral or clinical signs were observed at 1 or 4 hours post-dose in any rats receiving the vehicle. No significant neurobehavioral effects were observed in rats administered 25, 50 or 100 mg/kg doses at the 1-hour post-dose observation. Decreased activity (2/6 animals) and decreased abdominal tone (1–2 animals out of 6) were observed in a small number of animals treated with NV-CoV-2 at 4-hour post-dose observation.

Body temperature was not affected by the intravenous administration of NV-CoV-2 at 25 or 50 mg/kg at 1 or 4 hours post-dose when compared to the vehicle control group (Table 6). Body temperature, 1 hour after the administration of 100 mg/kg dose of NV-CoV-2 was

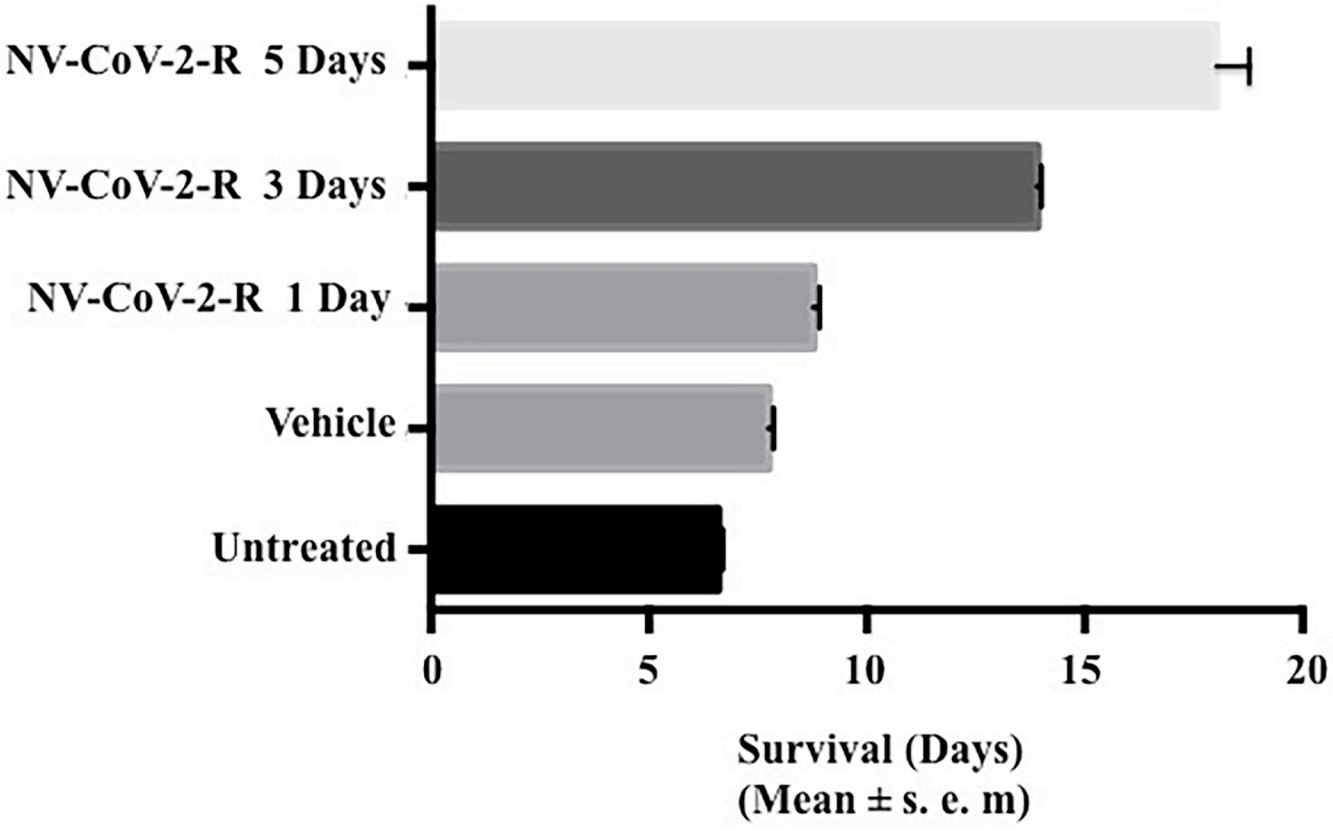

**Fig 5. Survival rate of NL-63 infected male rats after treatment with NV-CoV-2 encapsulated RDV (NV-CoV-2-R).**

slightly higher (p≤0.05) compared to the control group. Since the difference was marginal (<1.0°C), it may not be considered biologically relevant. Body temperature was not statistically different at 4 hours post-dose by the 100 mg/kg dose of NV-CoV-2.

## Cardiovascular parameter results

**Blood pressure and heart rate.** Random occurrence of fluctuations in blood pressure and heart rate are inherent in conscious freely moving animals. Hence consistent (2 or more time points) changes of 15% or more from vehicle controls were considered a significant (biologically relevant) change for blood pressure and heart rate. No significant effects on blood pressure (systolic, diastolic and mean) or heart rate were observed after the intravenous administration of 25, 37.5 or 50 mg/kg doses of NV-CoV-2 (data not shown).

**ECG.** All monkeys maintained sinus rhythm throughout the study. The intravenous administration of 25, 37.5 and 50 mg/kg NV-CoV-2 did not induce any significant effects on quantitative ECG parameters, as measured using the analysis software, in conscious telemetered cynomolgus monkeys.

**Immunogenicity.** This assay detects all relevant Immunoglobulin (Ig) isotypes. Since a non-mucosal route of administration was employed and there was no indication of anaphylaxis, the relevant isotypes would be IgG and IgM. The above screening assay is validated to determine lower limit of detection level of isotopes IgG and IgM as a group, and is scientifically justified for quantification of IgG and IgM as a group, using a linear calibration curve. This ELISA has a reliable lower limit of detection of 15ng/ml at which all relevant Ig Isotope's are

**Table 5. Effects of the vehicle and NV-CoV-2 on pulmonary parameters.**

| Intravenous Treatment | Time (hr.) | Respiratory Rate (breaths/min) | | Tidal Volume (mL) | | Minute Volume (mL/min) | |
|---|---|---|---|---|---|---|---|
| | | Mean | SEM | Mean | SEM | Mean | SEM |
| Vehicle Control (mL/kg) | 0 | 133.63 | 9.11 | 1.34 | 0.08 | 174.91 | 12.81 |
| | 0.25 | 137.72 | 7.99 | 1.53 | 0.10 | 203.24 | 9.20 |
| | 1 | 142.68 | 12.27 | 1.46 | 0.14 | 197.87 | 9.24 |
| | 2 | 142.84 | 13.18 | 1.36 | 0.11 | 185.35 | 8.79 |
| | 4 | 137.05 | 6.01 | 1.51 | 0.05 | 204.08 | 11.29 |
| NV-CoV-2 (25 mg/kg) | 0 | 142.09 | 10.23 | 1.45 | 0.09 | 199.60 | 9.67 |
| | 0.25 | 142.54 | 16.48 | 1.43 | 0.09 | 199.98 | 20.42 |
| | 1 | 150.46 | 9.23 | 1.27 | 0.06 | 185.69 | 10.27 |
| | 2 | 146.38 | 10.62 | 1.29 | 0.07 | 183.53 | 12.49 |
| | 4 | 139.34 | 7.66 | 1.34 | 0.07 | 183.37 | 10.95 |
| NV-CoV-2 (50 mg/kg) | 0 | 157.58 | 6.35 | 1.33 | 0.12 | 208.67 | 15.97 |
| | 0.25 | 151.88 | 5.56 | 1.43 | 0.08 | 211.23 | 11.82 |
| | 1 | 146.85 | 4.29 | 1.37 | 0.10 | 199.68 | 13.94 |
| | 2 | 146.07 | 8.10 | 1.40 | 0.10 | 201.93 | 20.20 |
| | 4 | 137.89 | 2.95 | 1.45 | 0.08 | 195.59 | 13.31 |
| NV-CoV-2 (100 mg/kg) | 0 | 126.70 | 10.66 | 1.45 | 0.14 | 189.56 | 26.62 |
| | 0.25 | 153.77 | 7.01 | 1.54 | 0.08 | 229.26 | 13.39 |
| | 1 | 152.22 | 10.75 | 1.50 | 0.11 | 220.26 | 11.19 |
| | 2 | 143.41 | 9.77 | 1.49 | 0.10 | 207.34 | 12.25 |
| | 4 | 137.21 | 10.93 | 1.45 | 0.08 | 195.24 | 19.10 |

Mean and standard error of the mean (SEM) calculated using non-truncated values. Time zero (0) = pre-dose value. There were no statistically significant (p≤0.05) changes compared to the vehicle control group.

**Table 6. Neurobehavioral parameters and body temperature.**

| Group No. | Treatment | Dose (mg/kg) | Volume (mL/kg) | Conc (mg/mL) | Signs Observed (15 Minutes-4 Hours post-dose[a]) | Rectal Body Temperature[b] (°C) | |
|---|---|---|---|---|---|---|---|
| | | | | | | 1 Hr. Post dose | 4 Hrs. post-dose |
| 1 | Vehicle control | 0 | 5 | 0 | No signs | 37.9 ±0.30 | 38.1 ±0.17 |
| 2 | NV-CoV-2 | 25 | 5 | 5 | 15 Min pd–No signs<br>1 Hr. pd—No signs<br>4 Hr. pd–Decreased activity (2/6), decreased abdominal tone (1/6) | 38.4 ±0.15 | 37.6 ±0.43 |
| 3 | NV-CoV-2 | 50 | 5 | 10 | 15 Min pd–No signs<br>1 Hr. pd—No signs<br>4 Hr. pd–Decreased activity (2/6), decreased abdominal tone (1/6) | 38.3 ± 0.16 | 38.0 ±0.25 |
| 4 | NV-CoV-2 | 100 | 5 | 20 | 15 Min pd–No signs<br>1 Hr. pd—No signs<br>4 Hr. pd–Decreased activity (2/6), decreased abdominal tone (2/6) | 38.8* ± 0.14 | 38.5 ± 0.16 |

[a] Observed for signs of pharmacological or toxicological activity at 15 minutes and 1 and 4 hours after dosing.

[b] Data are presented as mean ± standard error of the mean.

* Statistically significant (p≤0.05) change compared to the vehicle group–ANOVA followed by a Turkey HSD Multiple Comparison Test (Systat v.9.01).

clearly distinguishable from negative control, and quantifiable with acceptable accuracy and precision.

Immunogenicity determination was done on all animals (4/sex/group) in duplicate following I.V. dosing of NV-CoV-2 in rats with serum samples collected at days 14 and 28 days following the start of the 6 sequential doses over 10 day period. The mean OD of all dose groups is not statistically distinguishable from the vehicle group, both sexes in both Day 14 and Day 28 animals. Under the conditions of the assay method, IgG and IgM antibodies against NV-CoV-2 were not detected and, therefore NV-CoV-2 was not shown to be immunogenic.

### Safety pharmacology conclusion

The intravenous administration of NV-CoV-2 at doses of 25, 50 and 100 mg/kg did not affect respiratory function in rats. No significant neurobehavioral effects were observed in all dose groups of rats at the 1-hour post-dose observation. Decreased activity (2/6 animals) and decreased abdominal tone (1–2 animals out of 6) were observed in a small number of animals treated with NV-CoV-2 at 4-hour post-dose observation that was not dose-dependent. Body temperature in rats was not affected by the intravenous administration of NV-CoV-2 at 25, 50 and 100 mg/kg.

The intravenous administration of NV-CoV-2 at doses of 25, 37.5 and 50 mg/kg in conscious telemetered cynomolgus monkeys did not induce any significant, biologically relevant effects on heart rate, arterial blood pressure, cardiac rhythm, and ECG parameters. All monkeys maintained sinus rhythm throughout the study.

### Discussion

Remdesivir (GS-5734) is a broad-spectrum antiviral nucleotide prodrug that easily passes the cell membrane and is rapidly converted into its active triphosphate form [38]. The triphosphate (RDV-TP) derivative competes with endogenous adenosine triphosphate (ATP) for incorporation in elongating RNA strands and arrest virus replication in RNA viruses such as Ebola virus (EBOV) [39], MERS-CoV [40], SARS-CoV and SARS-CoV-2 [41]. For example, RDV effectively inhibits RNA viruses of the *Paramyxoviridae*, *Pneumoviridae* and *Coronaviridae* families with the $EC_{50}$ in the sub-micromolar range [42–44]. RDV also inhibits zoonotic and epidemic human coronaviruses [45]. Inhibitory effects of RDV on SARS-CoV-2 was first evaluated in cell culture studies using the Vero E6 African green monkey kidney cell lines. These cells express ACE2, which supports entry and replication of SARS-CoV-2 [46, 47]. RT-PCR assay showed a clinical virus isolated from Wuhan city was inhibited by RDV, with an $EC_{50}$ of 0.77 μM [48]. Similar results on the inhibitory effects of RDV were reported with SARS-CoV-2 isolated from Australian (VIC01/2020) [49] and Hong Kong (20001061/2020) [50].

RDV demonstrated beneficial therapeutic effects in several animal models of SARS- and MERS-CoV infections and in MERS-CoV-infected non-human primates [44, 51, 52]. A recent study demonstrated RDV activity was evaluated in a SARS-CoV-2 infected macaque model. Animals were treated 12 h post-infection with 10 mg/kg (day 1) followed by 5 mg/kg daily (day 2–6) of RDV, which is the equivalent dose recommended for humans [53]. When compared to the animal placebo control group, RDV diminished the clinical signs of the disease and reduced lung virus titers and tissue damage in all the treated animals [54].

RDV was evaluated in two randomized controlled clinical trials and the results have been published. The first trial was conducted in China and remained inconclusive due to the insufficient recruitment of patients [55]. The adaptive COVID-19 treatment trial treated 1063 patients with RDV, which showed a significant reduction in the recover time from 15 to 11

days. However, the reduced mortality rates from the treatment group were not statistically significant [56]. Wang *et al*. found no significant difference in the amount of virus RNA from the upper and lower respiratory tract specimen of patients treated with RDV or placebo [55].

The safety data concluded that RDV was well tolerated without any adverse side effects. However, in randomized clinical trials several hepatic drug-metabolizing enzymes like CYP2C8, CYP2D6 and CYP3A4 were observed elevated in both the RDV-treated and placebo groups. This might explain COVID-19-associated liver injury [57–60]. The FDA approved RDV in 2020 as the only therapeutic regimen to treat COVID-19 [57]. Besides, another drug LAGEVRIO™ (molnupiravir) though has not been approved by FDA yet, but has been authorized on December 23, 2021, for emergency use to treat severe COVID-19 symptoms and for whom alternative COVID-19 treatment options are not available [61].

RDV drug efficacy against COVID-19 does not correlate with the clinical outcomes in humans' despite the promising results of RDV *in vitro*. More importantly, RDV is not stable in presence of plasma, which may decrease the effective concentration of RDV *in vivo*. Our present study was designed to encapsulate RDV by using a platform technology based biopolymer comprised of a PEG-1000 (polyethylene glycol) and C16-alkyl pendants in the monomer unit [12, 62]. The PEG polymer forms the hydrophilic shell since the alkyl chains float together and forms a flexible core, like an immobilized oil droplet [9, 10]. A nanoviricide mimics a human cell and is large enough for a virus particle to latch onto it, yet small enough to circulate readily in the body. The purpose of the nanoviricide is to wrap around the virus particles and encapsulates it. Basically, we have developed a "Venus-FlyTrap" concept for virus particles. When a virus encounters our nanoviricide, it thinks it is binding to its target cell receptor(s), but the virus ends up getting engulfed and destroyed in the process (Fig 6A). This concept is quite

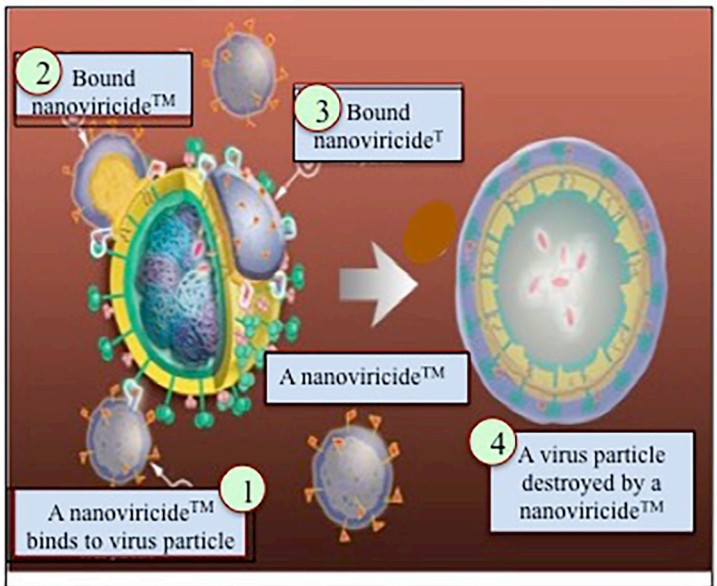
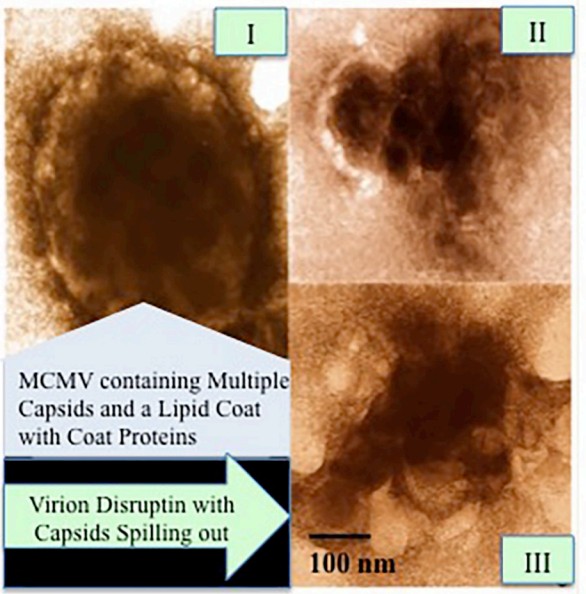

**Fig 6. Graphic representation of mechanism action of nanoviricide.**

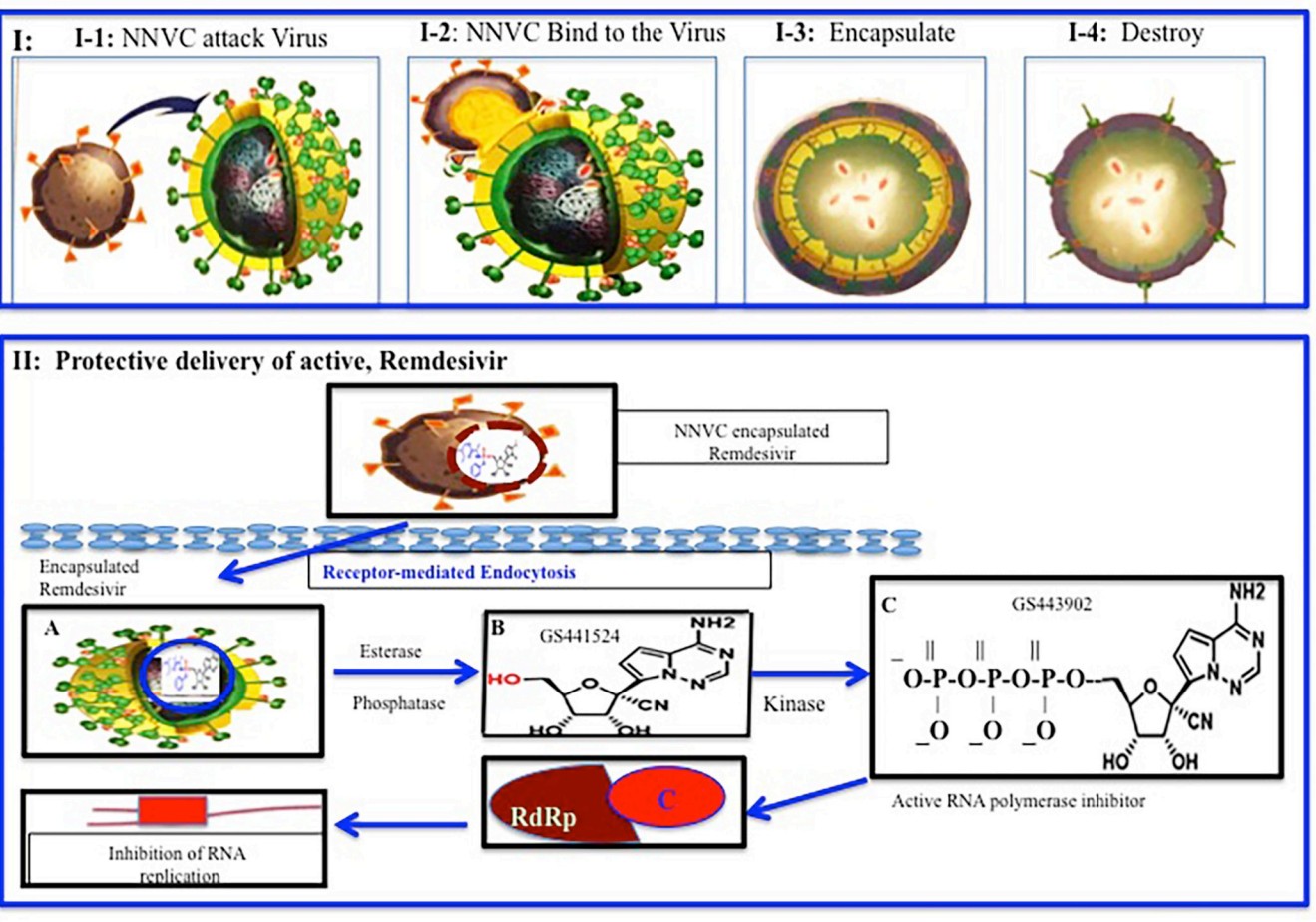

**Fig 7. Graphical representation of NV-CoV-2 action mechanism.**

similar to polymeric vs. monomeric inhibitors that has been shown in the scientific literature [9, 10]. By using this plug-and-play approach, the virus binding ligand portion of the nanoviricide could be altered and changed to attack different virus mutations that is beneficial for future use.

An analogy between nanoviricides and neutralizing anti-viral antibodies could be drawn. Antibodies can neutralize and cause aggregation on the virus particles, which is part of an immune response. They inhibit the endocytic uptake of viruses into the cells and/or prevents virus un-coating in endosomes. In addition, virus-antibody complexes can be destroyed by phagocytosis. A nanoviricide micelle will bind to a virus particle through van der Waals interactions and promotes encapsulation. This process may dismantle the viral envelope and prohibit the virus from entering the cell. This proposed mechanism of action is supported by electron photomicrographs (Fig 6B) of murine cytomegalovirus (CMV) incubated with a nanoviricide. It is evident that the inactivity of virus is due to the disruption of capsid organization (Fig 6B, Panels II and III).

The recent study demonstrated a very short life ($t_{1/2}$ = 5–8 min) of RDV in the presence of plasma. When RDV is encapsulated in our platform technology based biopolymer NV-387, it effectively becomes more stable ($t_{1/2}$ = 24h). RDV will eventually leak out slowly into the bloodstream from the polymeric nano-micelle, changing its protection against metabolism

over time. As a result, RDV encapsulated by our nanoviricide is an improved treatment of SARS_CoV-2 than the Gilead RDV alone. In addition, this study exhibited no evidence on any adverse reactions from our nanoviricides in any of the animals reported. This study was conducted under GLP-like conditions by AR BioSystems, Inc., Odessa, Tampa, FL. There are further microscopic histology and blood work analyses in progress [63, 64].

Our nanoviricide provides a safe cargo for transport through the plasma to deliver a small drug, like RDV to its target. RDV is the small drug approved by the FDA for treating COVID-19. RDV was found to be effective *in vitro* but didn't experienced similar effectiveness *in vivo* [65–67]. In addition, our nanoviricide has dual effect on the SARS-CoV-2 virus. First, it latches onto the virus, encapsulates it and disables the virus ability to infect their host cells. Secondly, it can encapsulate a small drug and protect it from the plasma during circulation in the body. As a result, our nanoviricide can effectively deliver a small drug to its target to inhibit virus proliferation (Fig 7).

## Conclusions

The current study showed a novel approach to improve the stability of RDV *in vivo*. RDV alone has a very short half-life in presence of plasma, which decreases the potency of RDV effects against SARS-CoV-2. The encapsulation of RDV in NV-387 polymer makes the drug highly stable in plasma. Even after overnight incubation with the plasma, the encapsulated RDV is comparable to the Gilead RDV. The researchers proposed that our nanoviricide possesses intrinsic antiviral activity and increases RDV effectiveness against SARS-CoV-2 by encapsulation. When our polymer encapsulates RDV, it protects the drug against plasma and allows it to get to its target. Potential mutations in the virus will occur but are very unlikely to enable our nanoviricide since its very versatile.

## Acknowledgments

We acknowledge all our colleagues, Secretaries for their help during the preparation of the manuscript by providing all the relevant information.

## Author Contributions

**Conceptualization:** Anil Diwan, Jay Tatake, Randall Barton.

**Data curation:** Vijetha Chiniga, Vinod Arora, Preetam Holkar, Yogesh Thakur, Jay Tatake.

**Formal analysis:** Anil Diwan, Vijetha Chiniga, Vinod Arora, Preetam Holkar, Jay Tatake, Neelam Holkar, Rajesh Pandey.

**Funding acquisition:** Anil Diwan.

**Investigation:** Ashok Chakraborty, Vijetha Chiniga, Preetam Holkar, Jay Tatake, Randall Barton, Rajesh Pandey, Bethany Pond.

**Methodology:** Ashok Chakraborty, Vijetha Chiniga, Vinod Arora, Preetam Holkar, Yogesh Thakur, Jay Tatake, Randall Barton, Neelam Holkar, Rajesh Pandey, Bethany Pond.

**Project administration:** Anil Diwan, Jay Tatake, Randall Barton, Neelam Holkar, Rajesh Pandey.

**Resources:** Ashok Chakraborty, Anil Diwan, Yogesh Thakur, Randall Barton, Neelam Holkar.

**Software:** Neelam Holkar, Bethany Pond.

**Supervision:** Ashok Chakraborty, Anil Diwan, Jay Tatake, Randall Barton.

**Validation:** Vinod Arora, Yogesh Thakur, Randall Barton.

**Visualization:** Jay Tatake, Randall Barton.

**Writing – original draft:** Ashok Chakraborty.

**Writing – review & editing:** Ashok Chakraborty, Anil Diwan, Bethany Pond.

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
