## [Decision Letter · Decision Letter 0]

10 Oct 2022

PONE-D-22-24874Dual Effects of NV-CoV-2 Biomimetic Polymer: An Antiviral Regimen Against COVID-19PLOS ONE

Dear Dr. Chakraborty,

Thank you for submitting your manuscript to PLOS ONE. After careful consideration, we feel that it has merit but does not fully meet PLOS ONE’s publication criteria as it currently stands. Therefore, we invite you to submit a revised version of the manuscript that addresses the points raised during the review process.

We look forward to receiving your revised manuscript.

Kind regards,

Syed Hani Hassan Abidi

Academic Editor

PLOS ONE

Reviewers' comments:

Reviewer's Responses to Questions

**Comments to the Author**

1. Is the manuscript technically sound, and do the data support the conclusions?

Reviewer #1: Yes

Reviewer #2: Yes

2. Has the statistical analysis been performed appropriately and rigorously? 

Reviewer #1: Yes

Reviewer #2: Yes

3. Have the authors made all data underlying the findings in their manuscript fully available?

Reviewer #1: Yes

Reviewer #2: Yes

4. Is the manuscript presented in an intelligible fashion and written in standard English?

Reviewer #1: Yes

Reviewer #2: Yes

5. Review Comments to the Author

Reviewer #1: Results needed to be explained better. They have made statement on the basis of unpublished a number of times. Why they have not shown data. Discussion did not discuss results in very effective manner. Discussion does not detailed similar studies. They have not compared their results with other studies.

Reviewer #2: The manuscript is well-written and well-presented. Besides, the study is well-designed. Only comment is that the authors claimed that “Remdesivir (RDV) is the only antiviral drug approved for COVID-19 therapy by the FDA”, indeed molnupiravir is also approved by FDA as anti-COVID-19 drug.

6. PLOS authors have the option to publish the peer review history of their article (what does this mean?). If published, this will include your full peer review and any attached files.

Reviewer #1: **Yes: **Muhammad Nouman Mughal

Reviewer #2: No

---

## [Author Response · Author response to Decision Letter 0]

19 Oct 2022

Rebuttal Letter: Oct 19, 2022

PLOS ONE Decision: Revision required [PONE-D-22-24874] -

Au: Done accordingly

Au: Data Availability statement: We want to reverse the statement what we had written earlier. No repository information is there that we have to submit later on. All the information are present in the manuscript.

Au: Orcid ID of Ashok Chakraborty (The first and Communicating Author) 

0000-0002-0232-3036

Au: “Data not shown”: Since those are not the core part of the Article, were removed.

Au: “Ethics statement” was deleted from the other section except the “Methods” section.

Reviewers' comments:

Reviewer's Responses to Questions

Comments to the Author

1. Is the manuscript technically sound, and do the data support the conclusions?

Reviewer #1: Yes

Reviewer #2: Yes

2. Has the statistical analysis been performed appropriately and rigorously?

Reviewer #1: Yes

Reviewer #2: Yes

3. Have the authors made all data underlying the findings in their manuscript fully available?

Reviewer #1: Yes

Reviewer #2: Yes

4. Is the manuscript presented in an intelligible fashion and written in standard English?

Reviewer #1: Yes

Reviewer #2: Yes

5. Review Comments to the Author

Reviewer #1: 

Q#1: Results needed to be explained better. They have made statement on the basis of unpublished a number of times. Why they have not shown data. 

Au: Those ‘data not shown’ part are the preliminary observation to standardize the materials and methods, and has little or no connection with the main paper, hence deleted from this revised version, also, the number of Figs remains within the limit.

Q#2: Discussion did not discuss results in very effective manner. Discussion does not detailed similar studies. They have not compared their results with other studies.

Au: Discussion part is now re-written accordingly, and English corrections were also done.

Reviewer #2: The manuscript is well-written and well-presented. Besides, the study is well-designed. Only comment is that the authors claimed that “Remdesivir (RDV) is the only antiviral drug approved for COVID-19 therapy by the FDA”, indeed molnupiravir is also approved by FDA as anti-COVID-19 drug.

Au: Thanks for updating with the new Information, and accordingly we have edited our Text (Please see, Lines 2-5); and also in the Discussion Section, Lines showed with Blue highlights

However, to exchange the information:

LAGEVRIO™ (molnupiravir) has not been approved, but has been authorized on December 23, 2021, for emergency use by FDA, for the treatment of mild-to-moderate COVID-19 in adults who are at high risk for progression to severe COVID-19, including hospitalization or death, and for whom alternative COVID-19 treatment options authorized by FDA are not accessible or clinically appropriate [https://www.lagevrio.com/patients/?utm]

6. PLOS authors have the option to publish the peer review history of their article (what does this mean?). If published, this will include your full peer review and any attached files.

Do you want your identity to be public for this peer review? For information about this choice, including consent withdrawal, please see our Privacy Policy.

Reviewer #1: Yes: Muhammad Nouman Mughal

Reviewer #2: No

---

## [Decision Letter · Decision Letter 1]

25 Nov 2022

Dual Effects of NV-CoV-2 Biomimetic Polymer: An Antiviral Regimen Against COVID-19

PONE-D-22-24874R1

Dear Dr. Chakraborty,

We’re pleased to inform you that your manuscript has been judged scientifically suitable for publication and will be formally accepted for publication once it meets all outstanding technical requirements.

Kind regards,

Syed Hani Hassan Abidi

Academic Editor

PLOS ONE

Additional Editor Comments (optional):

Reviewers' comments:

Reviewer's Responses to Questions

**Comments to the Author**

1. If the authors have adequately addressed your comments raised in a previous round of review and you feel that this manuscript is now acceptable for publication, you may indicate that here to bypass the “Comments to the Author” section, enter your conflict of interest statement in the “Confidential to Editor” section, and submit your "Accept" recommendation.

Reviewer #1: All comments have been addressed

Reviewer #2: All comments have been addressed

2. Is the manuscript technically sound, and do the data support the conclusions?

Reviewer #1: Yes

Reviewer #2: Yes

3. Has the statistical analysis been performed appropriately and rigorously? 

Reviewer #1: Yes

Reviewer #2: Yes

4. Have the authors made all data underlying the findings in their manuscript fully available?

Reviewer #1: Yes

Reviewer #2: Yes

5. Is the manuscript presented in an intelligible fashion and written in standard English?

Reviewer #1: Yes

Reviewer #2: Yes

6. Review Comments to the Author

Reviewer #1: (No Response)

Reviewer #2: (No Response)

7. PLOS authors have the option to publish the peer review history of their article (what does this mean?). If published, this will include your full peer review and any attached files.

Reviewer #1: No

Reviewer #2: No

---

## [Editor Report · Acceptance letter]

2 Dec 2022

PONE-D-22-24874R1 

Dual Effects of NV-CoV-2 Biomimetic Polymer: An Antiviral Regimen Against COVID-19 

Dear Dr. Chakraborty:

I'm pleased to inform you that your manuscript has been deemed suitable for publication in PLOS ONE. Congratulations! Your manuscript is now with our production department. 

Kind regards, 

on behalf of

Dr. Syed Hani Hassan Abidi 

Academic Editor

PLOS ONE